# Impact of COVID-19 on new pharmacotherapy for insomnia: A matched cohort study using the national insurance claims database in Japan

Daisuke Miyamori[1]*, Shuhei Yoshida[1], Wataru Omori[2], Saori Kashima[3,4], Masanori Ito[1]

1 Department of General Internal Medicine, Hiroshima University Hospital, Hiroshima, Japan,
2 Department of Psychiatry, NHO Kure Medical Center and Chugoku Cancer Center, Kure, Japan,
3 Center for the Planetary Health and Innovation Science, The IDEC Institute, Hiroshima University, Hiroshima, Japan, 4 Environmental Health Sciences Laboratory, Graduate School of Advanced Science and Engineering, Hiroshima University, Hiroshima, Japan

* morimiya@hiroshima-u.ac.jp

## Abstract

### Background

The COVID-19 pandemic has profoundly affected impacted both physical and mental well-being. This matched cohort study investigated the effects of COVID-19 on pharmacological treatments for insomnia, using Japan's National Insurance Claims Database.

### Methods

Data were matched by age, sex, Charlson Comorbidity Index (CCI) score, and enrollment month. Incidence rate ratios (IRR) and incidence rate differences (IRD) were calculated and compared for insomnia medication initiation, and subgroups based on age and sex categories. Sensitivity analyses were conducted for segmented intervals of 0–4, 5–12, and after 12 months.

### Results

The study included approximately 2 million pairs, predominantly women (59.4%), with a median follow-up of 7 months (Interquartile range, 4–12). Initiation of any insomnia medications occurred 77626 and 43142 times in the COVID-19 and control groups, respectively. IRR for new prescriptions was 1.7 times higher in the COVID-19 group (IRR: 1.71, 95% confidence interval [CI] 1.69–1.73), with an IRD of 1,634 events per 1,000,000 person-months (95%CI: 1599–1669). Non-Benzodiazepines and short-acting Benzodiazepines had the highest excess burdens among secondary end-points. The risk was observed in all age categories, even in under 20 years (younger individuals: IRR: 1.46 95%CI 1.39–1.53, IRD: 380, 95%CI 333–427). Sensitivity

**Data availability statement:** The data are owned by the Ministry of Health, Labor, and Welfare in Japan (MHLW). Access to the data requires approval from the MHLW. Requests for data access may be sent to the MHLW's data access committee at https://www.mhlw.go.jp/content/12400000/001158704.pdf.

**Funding:** This study was financially supported by the Innovative Research Program on Suicide Countermeasures (Grant Number JPSCIRS20220305) and JSPS KAKENHI Grant-in-Aid for Early-Career Scientists (Number 21K17227). The funders had no role in study design, data collection and analysis, decision to publish, or preparation of the manuscript.

**Competing interests:** The authors have declared that no competing interests exist.

analysis confirmed an increased risk over time, even after 12 months (IRD: 752, 95%CI 662–842).

## Conclusion

COVID-19 significantly associates with an elevated risk of insomnia medication initiation, emphasizing the necessity for mental health support in post-COVID-19 care. This study offers insights into the pandemic's influence on pharmacological treatment practices.

## Introduction

The COVID-19 pandemic has significantly impacted global health, causing both acute symptoms and a wide range of post-acute sequelae (PASC). [1–3] Beyond respiratory disease, the pandemic has profoundly influenced sleep patterns and insomnia rates. Several studies have reported increased prevalence of sleep disorders and insomnia during the pandemic [4]. However, trends in pharmacological treatment remains inconsistent. Given this uncertainty, it is crucial to investigate the potential impact of COVID-19 on insomnia treatment needs and hypnotic medication use.

Potential mechanisms for developing post-COVID insomnia include neuroinvasive potential of SARS-CoV-2, autonomic dysfunction, [5]. and, pandemic associated psychological stresses [6], [7]. Post-acute sequelae of SARS-CoV-2 infection (PASC) are persistent symptoms that can last for weeks or even months after initial infection with SARS-CoV-2. [8] These symptoms include fatigue, brain fog, shortness of breath, and other persistent symptoms that can significantly impacting the quality of life and overall health. [9–11] Factors such as male sex, high socioeconomic status, preexisting health conditions, and psychosocial factors may contribute to the development of PASC [12].

Japan presents a unique situation: until recently, cognitive-behavioral therapy for insomnia was seldom reimbursed or recommended, making pharmacotherapy the primary treatment option. [13,14] A cohort study on hospitalized patients showed an increased risk of in Japan; however, studies investigating the risk of initiation of pharmacological treatment for insomnia using population database are scarce. [15] The impact of COVID-19 on mental health, particularly insomnia, represents a potentially huge public health issue. Chronic insomnia is associated with increased risks of depression, anxiety, cardiovascular disease, and all-cause mortality [16,17]. Moreover, the economic burden of insomnia, including healthcare costs and lost productivity, is substantial [18]. Understanding the relationship between COVID-19 and subsequent insomnia medication use can inform public health strategies for post-pandemic mental health support and resource allocation.

This study aimed to evaluate the impact of COVID-19 on the initiation of pharmacological treatments for insomnia. We conducted a matched cohort study using the National Database of Health Insurance Claims and Specific Health Checkups of Japan (NDB). By examining demographic and clinical subgroups, we sought to

identify populations at highest risk for developing insomnia following COVID-19 infection, thereby informing targeted interventions and support strategies.

## Methods

### Research Design

This was a matched cohort study using the NDB.

### Data Source

This included outpatient and inpatient information stored at the individual level, which allowed us to track patient-based visits and treatment. The claims database included information on age, sex, diagnoses based on the International Statistics Classification of Diseases and Related Health Problems, Tenth Revision codes (ICD-10) with Japanese texts, and drugs dispensed based on the Anatomical Therapeutic Chemical (ATC) Classification System. The data for this research were accessed on 16/08/2024.

### Study Participants

Participants were individuals from five prefectures in Japan (Kyoto, Osaka, Hyogo, Okayama, and Hiroshima), who used the National Insurance Claims Database between January 2020 and December 2022. The area covered 20.5 million people and approximately 16% of Japan's population. In the five prefectures included in this study, the age-demographic composition was similar to that of the nation as a whole, and the cost frequency for psychiatric disorders as of 2021 did not show extreme bias [19] (S1 and S2 Tables).

### Outcome

The outcome of interest of this study was the initiation of prescription for insomnia which are available for outpatients in Japan, with insomnia being the main indication, and other indications such as anxiety and epilepsy being excluded. We categorize the drug as follows: Melatonin Receptor Agonist (MRA), Non-Benzodiazepine Hypnotic (non-BZO), benzodiazepine (short-acting [SA-BZO], intermediate-/long-acting [ILA-BZO]), Orexin Receptor Antagonist (ORA) and Barbiturates. The composite endpoint included the initiation of medication in these categories. The secondary endpoint was the new prescription for each drug category. A detailed list of the medications included in each category is provided in S3 Table.

### Exposure

Subjects diagnosed with COVID-19 during the study period were assigned to the COVID-19 group. The government exempted individuals diagnosed with COVID-19 during the observation period from incurring medical expenses. In this study, the public expense number on payment was used to determine the exposure status [20].

### Inclusion and exclusion criteria

All individuals who were insured and had access to healthcare from January 2020, when COVID-19 was prevalent, to December 2022, were included in this study. We excluded those who did not have a 1-year look-back period before the index month. We also excluded those who had received hypnotics drugs before the index month during the 1-year pre-assessment period. Details of the data assessment in this study are provided in S1 Fig.

### Data collection

The NDB served as the primary data source, providing details such as participant's age, sex, and ICD-10 codes for registered diseases and information on their prescription history, including specific drug names. Japan's universal health

insurance system features a detailed NDB that thoroughly records each outpatient appointment, hospital stay, diagnostic code, and medication prescribed for all residents. The Charlson Comorbidity Index (CCI) was determined utilizing the ICD-10 codes derived from the registered disease codes prior to the index month for both the infected and control groups. [21,22]

### Data Matching

The data organization was based on four variables: age category, sex, CCI total score, and month of enrollment in the insurance claims database. The month the participants were infected with COVID-19 was designated the index month. To ensure an accurate matching algorithm, we sought controls from the claims database that perfectly matched the cases for all four variables. For each month, controls matched on all four criteria were identified and paired randomly with cases. Once matched, participants were used exclusively once and were not reused in subsequent matching. The validity of the matching was assessed by comparing comorbidities between groups using standardized mean differences (SMD), with an SMD ≤ 0.1 considered indicative of balanced groups.

### Follow-up of the cohorts

The maximum follow-up period in this study was 24 months. All individuals in the matched cohort were tracked from the index date to the initiation of new pharmacotherapy for insomnia or the conclusion of the follow-up period, whichever occurred first.

### Statistical analyses

We used the Kaplan–Meier estimator to calculate the cumulative incidence of the composite endpoint that occurred over time, starting from the index date. Subsequently, we assessed the incidence rate difference (IRD) and incidence rate ratio (IRR) per 1,000,000 person-months between the COVID-19 and control groups, along with 95% confidence intervals (CI). For subgroup analysis, we calculated the IRD and IRR of the composite outcomes between the COVID-19 and Control groups for subgroups of age and sex categories. In addition, a sensitivity analysis was conducted to assess the robustness of the main and subgroup analyses by splitting the study period into three intervals: the first 4 months, between 5 and 12 months, and after 12 months since the COVID-19 infection or index month.

### Ethical consideration

The study protocol was reviewed and approved by the Epidemiological Research Committee of Hiroshima University (approval number: E2022-0024-01). The data for this research were owned by Ministry of Health, Labor, and Welfare in Japan (MHLW) and obtained by the authors with ethical approval (approval number: 1502).

## Results

Fig 1 shows the flowchart of this study. This study included approximately 16 million participants, with 3 million individuals infected with COVID-19. After excluding individuals who did not have a 1-year look-back period and individuals who had already received prescriptions for insomnia, A COVID-19 group of 2375621 individuals was matched with a non-infected group. After matching for age category, gender, CCI, and insurance enrollment month, a total of 2,226,589 pairs were included in the study.

Table 1 presents the baseline characteristics of 4,453,178 study participants, equally divided between the control group and the COVID-19 group (2,226,589 each). The table shows demographic information and comorbidities for both groups. The standardized mean difference (SMD) is used to assess the balance between groups, with values close to zero indicating good balance. The participants were predominantly women (54.94%), and the largest age category was the 70s.

Fig 2 shows the Kaplan–Meier curves for participants infected with COVID-19 and matched controls for all participants, and Table 2 shows the IRR and IRD for the composite endpoint and secondary endpoint. The cumulative incidence of

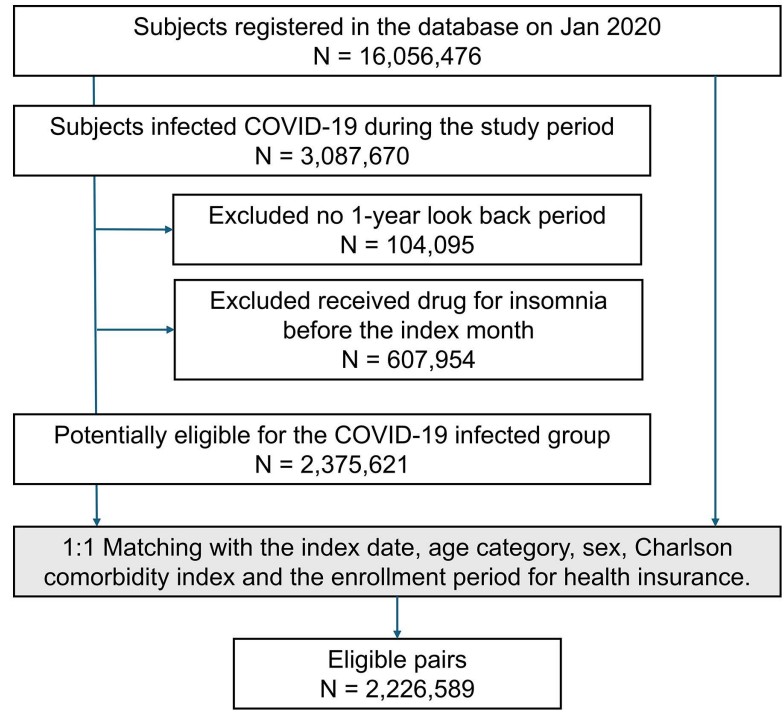

**Fig 1. Flow chart of Study Participants.** Among roughly 16 million out of an estimated 3 million individuals, approximately 16 million were infected. After excluding 607,954 individuals who had received hypnotics medication within the past year, 2,375,621 subjects were eligible for the COVID-19 group and the matched cohort study. Lastly, 2,226,589 pairs were matched and analyzed. In the matching process, age categories were divided into groups every 1 year for subjects under 10 years old, and in increments of 5 years for subjects aged 10 years and older. The Charlson Comorbidity Index was matched using total scores.

pharmacotherapy for insomnia continuously increased during the study period. The included 2.2 million pairs were followed for a median duration of 7 months (Interquartile range 4–12). Of the 2,226,589 individuals in each group, 77,626 and 43,142 cases of the composite endpoint occurred in the COVID-19 and control groups, respectively, over a 24-month observational period. The IRR was 1.71 (95% CI, 1.69–1.73) and IRD was 1,634 events per 1,000,000 person-months (95% CI, 1599–1669). The analyses of secondary endpoints showed an increased risk of prescription for any of the hypnotic categories in the COVID-19 group compared with the control group. During the study period, ORA and non-BZO were the most frequently prescribed, and the IRD was 791 (95% CI: 767–815) and 614 (95% CI: 592–636) events per 1,000,000 person-months, and the IRR was 1.73 (95% CI: 1.70–1.76) and 1.70 (95% CI: 1.67–1.82), respectively. In the subgroup analyses, the risk of new prescriptions significantly increased across categories.

Fig 3 shows the IRR and IRD for the subgroups; the numbers are shown in S4 Table. The IRR and IRD were elevated in all subgroups, consistent with the main analysis.

The sensitivity analysis presented in Fig 4 shows the incidence risk ratios and differences for the composite endpoint across the overall and subgroup categories, into three distinct time intervals: within 4 months, between 5 and 12 months, and after 1 year; the numbers are shown in S5 Table. The incidence risk ratios within 4 months, between 5 and 12 months, and after 12 months were 2.33 (95% CI: 2.28–2.37), 1.46 (95% CI: 1.43–1.48) and 1.25 (95% CI: 1.21–1.28), respectively. The incidence rate of new prescriptions for insomnia was higher in the COVID-19 group than in the control group for all subgroup categories, even after the 1-year follow-up period. Sensitivity analyses indicated that the frequency exhibited a declining trend over time following infection, yet remained significantly higher than that in the control group.

**Table 1. Baseline characteristics of study participants.**

| | Total | Control group | COVID-19 group | SMD |
|---|---|---|---|---|
| | N = 4,453,178 | N = 2,226,589 | N = 2,226,589 | |
| Women (%) | 2,443,966 (54.9%) | 1,221,983 (54.9%) | 1,221,983 (54.9%) | 0 |
| Age category (%) | | | | 0 |
| 0–4 years | 315,238 (7%) | 157,619 (7%) | 157,619 (7%) | |
| 5–9 years | 259,698 (6%) | 129,849 (6%) | 129,849 (6%) | |
| 10–14 years | 186,020 (4%) | 93,010 (4%) | 93,010 (4%) | |
| 15–19 years | 209,464 (5%) | 104,732 (5%) | 104,732 (5%) | |
| 20–24 years | 190,654 (4%) | 95,327 (4%) | 95,327 (4%) | |
| 25–29 years | 203,532 (5%) | 101,766 (5%) | 101,766 (5%) | |
| 30–34 years | 216,556 (5%) | 108,278 (5%) | 108,278 (5%) | |
| 35–39 years | 217,038 (5%) | 108,519 (5%) | 108,519 (5%) | |
| 40–44 years | 205,936 (5%) | 102,968 (5%) | 102,968 (5%) | |
| 45–49 years | 258,474 (6%) | 129,237 (6%) | 129,237 (6%) | |
| 50–54 years | 266,552 (6%) | 133,276 (6%) | 133,276 (6%) | |
| 55–59 years | 231,032 (5%) | 115,516 (5%) | 115,516 (5%) | |
| 60–64 years | 229,702 (5%) | 114,851 (5%) | 114,851 (5%) | |
| 65–69 years | 239,540 (5%) | 119,770 (5%) | 119,770 (5%) | |
| 70–74 years | 433,070 (10%) | 216,535 (10%) | 216,535 (10%) | |
| 75–79 years | 285,654 (6%) | 142,827 (6%) | 142,827 (6%) | |
| 80–84 years | 277,520 (6%) | 138,760 (6%) | 138,760 (6%) | |
| 85 + years | 227,498 (5%) | 113,749 (5%) | 113,749 (5%) | |
| CCI (%) | | | | 0 |
| 0 | 2,316,576 (52%) | 1,158,288 (52%) | 1,158,288 (52%) | |
| 1 | 879,556 (20%) | 439,778 (20%) | 439,778 (20%) | |
| 2–3 | 740,272 (17%) | 370,136 (17%) | 370,136 (17%) | |
| 4 or above | 516,774 (12%) | 258,387 (12%) | 258,387 (12%) | |
| Comorbidities | | | | |
| AMI | 66,427 (1.5%) | 33,129 (1.5%) | 33,298 (1.5%) | 0.001 |
| CHF | 462,268 (10.4%) | 223,463 (10.0%) | 238,805 (10.7%) | 0.02 |
| PVD | 410,040 (9.2%) | 203,022 (9.1%) | 207,018 (9.3%) | 0.01 |
| CEVD | 493,194 (11.1%) | 242,035 (10.9%) | 251,159 (11.3%) | 0.01 |
| Dementia | 90,640 (2.0%) | 36,047 (1.6%) | 54,593 (2.5%) | 0.06 |
| CPD | 339,288 (7.6%) | 161,171 (7.2%) | 178,117 (8.0%) | 0.03 |
| Rheumatoid | 139,994 (3.1%) | 68,156 (3.1%) | 71,838 (3.2%) | 0.01 |
| PUD | 849,366 (19.1%) | 413,469 (18.6%) | 435,897 (19.6%) | 0.03 |
| Diabetes | 271,083 (6.1%) | 138,150 (6.2%) | 132,933 (6.0%) | −0.01 |
| Liver disease | 857,180 (19.2%) | 430,785 (19.3%) | 426,395 (19.2%) | −0.01 |
| Cancer | 361,892 (8.1%) | 185,429 (8.3%) | 176,463 (7.9%) | −0.01 |
| HP/PAPL | 38,807 (0.9%) | 18,656 (0.8%) | 20,151 (0.9%) | 0.01 |
| RD | 134,674 (3.0%) | 63,942 (2.9%) | 70,732 (3.2%) | 0.02 |
| AIDS | 1,970 (0.0%) | 1,078 (0.0%) | 892 (0.0%) | −0.004 |

CCI: Charlson comorbidity index. AMI, acute myocardial infarction; CHF, congestive heart failure; PVD, peripheral vascular disease; CEVD, cerebrovascular disease; Dementia, dementia; CPD, chronic pulmonary disease; Rheumatoid, rheumatoid and other connective tissue diseases (including collagen vascular diseases); PUD, peptic ulcer disease; Diabetes, diabetes mellitus; Liver disease, chronic liver disease; Cancer, malignant neoplasm; HP/PAPL, hemiplegia/paraplegia; RD, renal disease; AIDS, acquired immune deficiency syndrome.

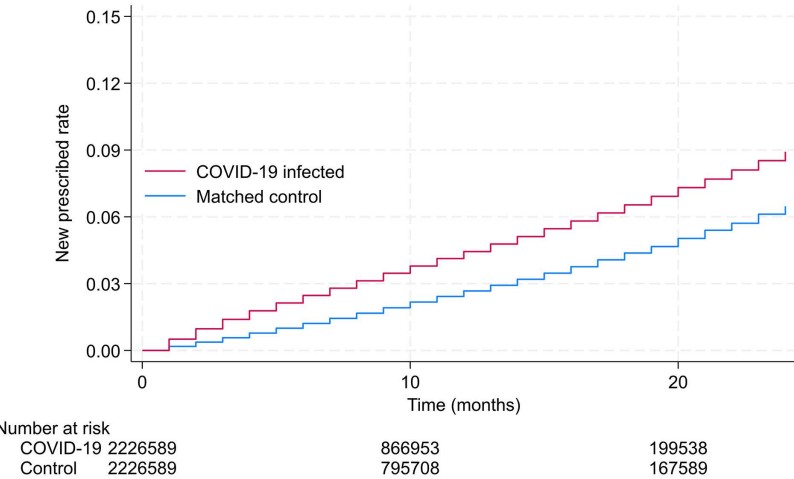

**Fig 2. Kaplan-Meier Analysis of New Prescription Rates for Insomnia.** Cumulative incidence of new pharmacotherapy for insomnia among individuals with COVID-19 (blue line) and a matched non-infected control group (red line). The p-value for the log-rank test was < 0.001, indicating a statistically significant difference between the two groups. The shaded areas around the lines represent 95% confidence intervals.

**Table 2. Cumulative Incidences of Pharmacotherapy for Insomnia after the COVID-19 infection in the Matched Cohort Design for Composite endpoint, secondary endpoint and subgroup categories.**

| | No. of pairs | No. of Events in COVID-19 Group | No. of Events in Control Group | Cumulative Incidence (No. of Events per 1 000 000 Person months) | |
| --- | --- | --- | --- | --- | --- |
| | | | | *Ratio (95% CI)* | *Difference (95% CI)* |
| Composite endpoint | 2226589 | 77626 | 43142 | 1.71 (1.69–1.73) | 1634 (1599–1669) |
| Secondary endpoints | | | | | |
| SA-BZO | 2226589 | 15945 | 8560 | 1.75 (1.71–1.80) | 341 (326–357) |
| ILA-BZO | 2226589 | 4263 | 2309 | 1.73 (1.65–1.82) | 90 (81–98) |
| Non-BZO | 2226589 | 29860 | 16585 | 1.70 (1.67–1.73) | 614 (592–636) |
| MRA | 2226589 | 12022 | 5708 | 1.98 (1.92–2.04) | 296 (283–310) |
| ORA | 2226589 | 37341 | 20324 | 1.73 (1.70–1.76) | 791 (767–815) |
| Barbiturates | 2226589 | 48 | 15 | 3.00 (1.65 – 5.77) | 1.59 (0.80 to 2.37) |

No.: number; CI: confidence interval; MRA: melatonin receptor agonist; non-BZO: Non-Benzodiazepine Hypnotics; SA-BZO: short-acting benzodiazepines; ILA-BZO: Intermediate/long-acting benzodiazepines; ORA: orexin receptor agonist.

## Discussion

This study aimed to investigate the incidence of initiating pharmacological treatments for insomnia after COVID-19 and compare it with that in individuals without infections. Our findings indicate that the incidence of initiating insomnia medications increased approximately 1.7 times in individuals with post-COVID-19 infection compared with those without infections. Subgroup analyses showed that this increased incidence was consistently observed across all age groups, including children and adolescents, and in both women and men (Fig 3 and S4 Table). The increased incidence of new insomnia medications persisted throughout the study period, even after 1-year of infection.

Studies reported a sharp rise in hypnotics after COVID-19, with an increase of 0.23 boxes per 1000 persons, [4] and higher insomnia diagnosis risk of 1.92 times at six months and 2.46 times at one year. [23] Additionally, 30% of severe

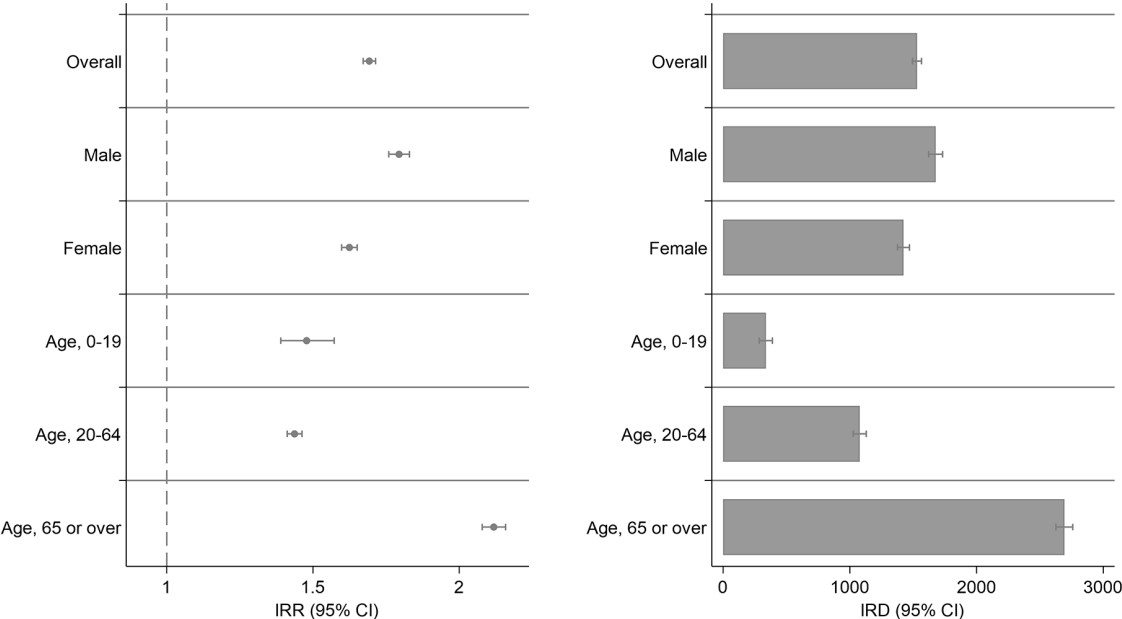

**Fig 3. Subgroup analysis for incidence rate difference and ratio.** This figure presents the Incidence Rate Ratios (IRR) with 95% Confidence Intervals (CI) on the left panel and the Incidence Rate Differences (IRD) with 95% Confidence Intervals (CI) on the right panel, stratified by demographic and clinical subgroups. Left Panel (IRR, 95% CI): The IRR compares the incidence rates between the exposure group and the control group within each subgroup. An IRR greater than 1 indicates a higher incidence rate in the exposure group compared to the control group. Right Panel (IRD, 95% CI): The IRD shows the absolute difference in incidence rates per 1,000,000 individuals between the exposure and control groups within each subgroup. A positive IRD indicates a higher incidence in the exposure group. The subgroups are the same as in the left panel.

PASC patients needed sleep-inducing drugs long-term. [24] Using the Population database, this study found the risk after COVID-19 persisted beyond 2 years. In Japan, cognitive-behavioral therapy for insomnia (CBT-I) is not covered by insurance, and pharmacotherapy remains the predominant treatment. [13] Therefore, the prescription of hypnotics in this study may reflects the onset of new cases of insomnia.

Insomnia exacerbates physical and psychiatric disorders, often with anxiety, depression, and increased stress. Chronic insomnia is linked to cognitive decline, reduced quality of life, and higher risks of injuries, suicidal thoughts, cardiovascular issues, diabetes, and cancer. [25–28] In addition, the mechanism of insomnia in PASC involves multiple biological and psychological factors. [21,29,30] Insomnia in PASC patients likely results from neuroinflammatory processes, autonomic dysfunction, direct viral effects on the central nervous system, psychological stressors, circadian rhythm disruptions, and physical symptoms. [31] Furthermore, the substantial increase in prescriptions of orexin receptor antagonists and non-benzodiazepine hypnotics observed in this study may have important implications for healthcare resource use, underscoring the need to expand access to cost-effective non-pharmacological interventions in Japan. The COVID-19 pandemic heightened challenges, impacting all generations and increasing healthcare burdens and economic losses due to reduced productivity. [32]

## Strengths and limitations

A key strength of this study is that it used a national insurance claims database, including 2.2 million COVID-19 and control individuals. Therefore, a comprehensive evaluation of personal-level risk factors for insomnia medication can be accomplished by utilizing a substantial, well-matched cohort database, which minimizes bias and presents more precise outcomes. Second, the certainty of the confirmation of COVID-19 infection would be accurate. Because the

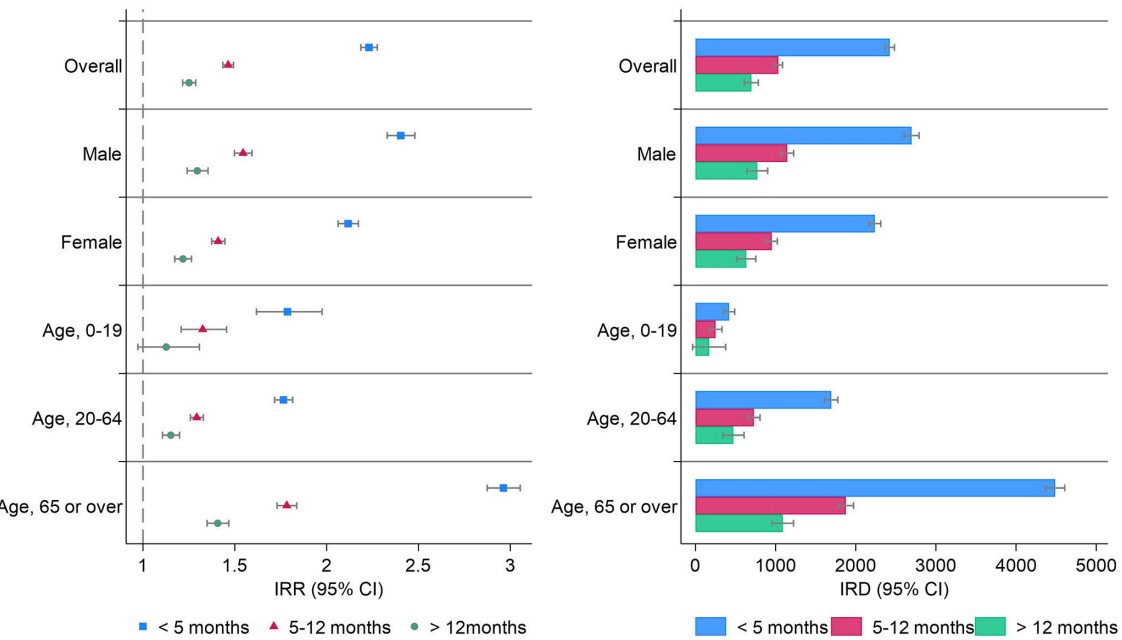

**Fig 4. Sensitivity analysis for incidence rate difference and ratio across segmented intervals.** This graph illustrates the sensitivity analysis of the incidence rate difference (IRD) and incidence rate ratio (IRR) across various categories based on age, sex, and the Charlson Comorbidity Index. The analysis is divided into four distinct time intervals: 0-4 months (Blue), 5–12 months (Red), and more than 12 months (Green). Left panel (IRR): The markers in the left panel represent the IRR across the time intervals, illustrating the relative risk of the incidence rate between the exposed and unexposed groups within each category. Right panel (IRD): The height of the bars in the right panel indicates the IRD, representing the absolute difference in incidence rates between the groups within each time interval. The vertical lines attached to the bars and plots represent the 95% confidence intervals for IRD and IRR. IRR, incidence rate ratio; IRD, incidence rate difference; CCI, Charlson comorbidity index; CI, confidence interval.

reimbursement for COVID-19 infection is paid by public funds, all insurance codes for the diagnosis and treatment of patients diagnosed with COVID-19 were recorded. Moreover, during the study period, all COVID-19 infections were mandatorily reported in Japan.

This study has limitations. First, our study lacked data on COVID-19 severity, which may have influenced outcomes. Second, selection bias may exist due to misclassification of COVID-19 infection and outcome status. Individuals with mild COVID-19 symptoms who were not diagnosed might have been classified as controls, although this bias might work as a point estimate into null. Insomnia may be more frequently prescribed in patients with post-COVID-19 conditions, potentially amplifying results. Third, we did not account for vaccination, economic, and employment statuses. Fourth, the codes identified have not undergone validation. Research assessing Japanese diagnostic codes for hospitalized patients showed sensitivity and specificity of 78.9% and 93.2%, respectively. [33]. Fifth, despite matching on key variables, the study may be subject to residual confounding from unmeasured factors, such as health seeking behaviors. A potential limitation of this study is the increased likelihood of healthcare-seeking behavior among individuals with COVID-19 compared to those without the disease. [34,35] The emphasis on early detection and isolation in public health messaging may have led to higher rates of medical attention-seeking. [35] These factors could have influenced the observed patterns in healthcare utilization and should be considered when interpreting the results. Sixth, this study sampled patients in only five prefectures, not all of Japan, and COVID-affected and insomnia patient rates may differ across prefectures. Similarly, generalizability is limited in countries with different cultural backgrounds and treatment policies. Finally, the possibility that medications were prescribed for reasons other than insomnia cannot be excluded entirely. [36,37] Future research should consider multi-country cohort studies to explore the generalizability of our findings across different healthcare systems

and cultural contexts. Longitudinal research with extended follow-up periods is necessary to determine the persistence of insomnia symptoms in patients post-COVID-19 and assess the efficacy and safety of pharmacotherapeutic interventions.

## Conclusions

In conclusion, our findings demonstrated a significant association between COVID-19 and a heightened risk of initiating insomnia pharmacotherapy. This underscores the urgent need to integrate mental health support into the post-COVID-19 care protocols. Ensuring timely intervention is crucial in managing mental health issues in the context of COVID-19. Our study contributes to a more detailed and contextual understanding of the impact of COVID-19 on pharmacological treatment practices.

Table of contents: This study on the impact of COVID-19 on new insomnia pharmacotherapy in Japan, using a matched cohort from the National Insurance Claims Database, revealed a significant 1.7-fold increase in insomnia medication initiation among COVID-19 patients compared with that in controls. Older individuals and those with higher Charlson Comorbidity Index scores are at high risk; the elevated risk persists beyond one year post-infection. This underscores the pandemic's impact on mental health and the need for integrated support in post-COVID-19 care.

## Supporting information

**S1 Table. Population of 5 Prefectures and Japan by Age Group and Sex.**
(DOCX)

**S2 Table. Average per capita medical expenses related to mental illness in targeted prefectures.**
(DOCX)

**S3 Table. ATC code, categories and drug names included in this study.**
(DOCX)

**S4 Table. Incidence Rate Ratios and Differences of Pharmacotherapy for Insomnia after the COVID-19 infection in the Matched Cohort Design for Composite endpoint and subgroup categories.**
(DOCX)

**S5 Table. Sensitivity Analysis of Pharmacotherapy for Insomnia after the COVID-19 infection in the Matched Cohort Design for Composite endpoint and subgroup categories.**
(DOCX)

**S1 Fig. Illustration of this study design.**
(DOCX)

## Author contributions

**Conceptualization:** Daisuke Miyamori, Shuhei Yoshida.

**Data curation:** Daisuke Miyamori, Shuhei Yoshida.

**Formal analysis:** Daisuke Miyamori.

**Funding acquisition:** Daisuke Miyamori.

**Investigation:** Daisuke Miyamori.

**Methodology:** Daisuke Miyamori.

**Resources:** Daisuke Miyamori.

**Software:** Daisuke Miyamori.

**Supervision:** Masanori Ito.

**Validation:** Wataru Omori.

**Visualization:** Daisuke Miyamori.

**Writing – original draft:** Daisuke Miyamori.

**Writing – review & editing:** Shuhei Yoshida, Wataru Omori, Saori Kashima.

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
