## [Decision Letter · Decision Letter 0]

19 Mar 2025

Dear Dr. Miyamori,

Thank you for submitting your manuscript to PLOS ONE. After careful consideration, we feel that it has merit but does not fully meet PLOS ONE’s publication criteria as it currently stands. Therefore, we invite you to submit a revised version of the manuscript that addresses the points raised during the review process.

We look forward to receiving your revised manuscript.

Kind regards,

Tatsuya Noda

Academic Editor

PLOS ONE

Journal Requirements:

2. Thank you for stating the following financial disclosure: This study was financially supported by the Innovative Research Program on Suicide Countermeasures (Grant Number JPSCIRS20220305) and JSPS KAKENHI Grant-in-Aid for Early-Career Scientists (Number 21K17227).

Additional Editor Comments:

Thank you for your submission.

I believe that the reviewers' comments are all important points.

The authors should adequately respond to all of the reviewers' comments before reaching a decision on publication.

As an editor, I consider it necessary to revise some of the analyses.

I hope that an improved manuscript will be resubmitted.

Reviewers' comments:

Reviewer's Responses to Questions

**Comments to the Author**

1. Is the manuscript technically sound, and do the data support the conclusions?

Reviewer #1: Partly

Reviewer #2: Partly

2. Has the statistical analysis been performed appropriately and rigorously?

Reviewer #1: I Don't Know

Reviewer #2: No

3. Have the authors made all data underlying the findings in their manuscript fully available?

Reviewer #1: Yes

Reviewer #2: No

4. Is the manuscript presented in an intelligible fashion and written in standard English?

Reviewer #1: Yes

Reviewer #2: No

Reviewer #1: The authors of this manuscript investigated the incidence of initiating pharmacological treatments for insomnia after COVID-19. I have several comments for this manuscript.

1. The authors need to cite appropriate previous studies and explain the scientific rationale on incidence of insomnia increase versus post-COVID-19 infection is so interesting. And then, authors need to support their study approach indicating the incidence of insomnia medications increase in individuals with post-COVID-19 infection can be reliable to measure that research question. However, the authors fail to explain the impact toward public health on COVID-19 infection and mental health which may be potentially a huge issue. These related statements remain fragile.

2. NDB is a medical claim database. Claimed disease name in these medical insurance databases is often different from diagnosed disease. Seems related validation study is not available and/or prepared for this study topic. The authors need to be cautious on how the data became generated. The authors need to explain why this study approach can reasonably identify the study patients with insomnia and COVID-19 infection.

3. Study patients are only from 5 prefectures; however, using NDB dataset. Please provide information related to methodologies that other researchers can reproduce this result. Furthermore, are these populations representable of nationwide Japan? Why did you choose these 5 prefectures instead of 47? I think regions with higher incidence rate and prevalence of mental health diseases are missing from this study population.

4. Table 1: Was the total study patients (N=4.4 million) reasonable level for explaining the incidence/ prevalence of insomnia among these 5 prefectures?

5. The authors need to provide scientifically reasonable information on why the outcomes are defined appropriately in this study settings since claimed disease and actual diagnosed disease are often different.

6. The authors need to provide scientifically reasonable information on why the prescriptions were made only for insomnia in this study settings and made it possible to exclude other indications.

7. The authors calculated CCI scores based on claimed insurance disease which is different from actual diagnosed disease. Please cite appropriate previous studies and explain how this approach is scientifically reasonable and reliable.

Reviewer #2: Thank you for the opportunity to review the manuscript. I believe that several points need to be addressed.

Major issues:

1. The authors should address the potential for differential misclassification bias related to insomnia. It is plausible that the majority of patients do not receive treatment for insomnia, and individuals in the exposure group may be more likely to receive such treatment compared to those in the control group, potentially due to differences in help-seeking behavior.

2. The authors concluded that the initiation of insomnia medication highlights the necessity of mental health support. However, clinical guidelines for the treatment of insomnia generally recommend nonpharmacological interventions as the first-line approach. Therefore, the initiation of pharmacological treatment for insomnia does not necessarily indicate a need for mental health support, but may instead reflect potentially inappropriate prescribing practices.

3. The authors stated that the rationale for this study was to investigate the long-term effects of COVID-19 on mental health. However, the median follow-up period was only seven months. The authors should clarify the justification for this follow-up duration in the context of evaluating long-term outcomes.

4. The authors identified insomnia medications using ATC codes. For transparency and reproducibility, please provide a complete list of the included drug names, their routes of administration, and corresponding domestic drug codes. In particular, clarification is needed on the following points: What is 'lilmazepam'?; Why were 'rilmazafone' and 'melatonin' not included?; How was 'haloxazolam' identified, given that it does not have an ATC code?; How were 'midazolam' and 'haloxazolam' excluded from the ATC category NC05CD?; Was flunitrazepam injection included in the analysis?; Why were other insomnia medications, such as barbiturates, not considered?

**Do you want your identity to be public for this peer review?** For information about this choice, including consent withdrawal, please see our Privacy Policy

Reviewer #1: No

Reviewer #2: No

---

## [Author Response · Author response to Decision Letter 1]

24 Apr 2025

PONE-D-25-01134

Impact of COVID-19 on New Pharmacotherapy for Insomnia: A Matched Cohort Study using the National Insurance Claims Database in Japan

PLOS ONE

Dear Dr. Miyamori,

Thank you for submitting your manuscript to PLOS ONE. After careful consideration, we feel that it has merit but does not fully meet PLOS ONE’s publication criteria as it currently stands. Therefore, we invite you to submit a revised version of the manuscript that addresses the points raised during the review process.

We look forward to receiving your revised manuscript.

Kind regards,

Tatsuya Noda

Academic Editor

PLOS ONE

Journal Requirements:

Thank you for your suggestions regarding the requirements for PLOS ONE. We have formatted the manuscript as you suggested.

2. Thank you for stating the following financial disclosure: This study was financially supported by the Innovative Research Program on Suicide Countermeasures (Grant Number JPSCIRS20220305) and JSPS KAKENHI Grant-in-Aid for Early-Career Scientists (Number 21K17227).

Thank you for your suggestions regarding the funder roles on the study. We have added the sentence as follows.

Funding, 2nd sentence.

Thank you for your suggestions regarding the data availability statement. The data are owned by Ministry of Health, Lavour and Welfare in Japan and needs permission to use. Therefore, we have added the sentence as follows.

Ethical Consideration and Data Availability section, 2nd sentence.

Data are owned by Ministry of Health, Labor, and Welfare in Japan (MHLW).

Thank you for your suggestions regarding the Supplemental data. We have added the captions for my supporting information at the end of the manuscript as follows.

Supporting Information

Supplementary Table 1. ATC code, categories and drug names included in this study.

Supplementary Table 2. Cumulative Incidences of Pharmacotherapy for Insomnia after the COVID-19 infection in the Matched Cohort Design for Composite endpoint, secondary endpoint and subgroup categories.

Supplementary Table 3. Cumulative Incidences of Pharmacotherapy for Insomnia after the COVID-19 infection in the Matched Cohort Design for Composite endpoint, secondary endpoint and subgroup categories.

Supplementary Figure 1. Illustration of this study design

Additional Editor Comments:

Thank you for your submission.

I believe that the reviewers' comments are all important points.

The authors should adequately respond to all of the reviewers' comments before reaching a decision on publication.

As an editor, I consider it necessary to revise some of the analyses.

I hope that an improved manuscript will be resubmitted.

Reviewers' comments:

5. Review Comments to the Author

Reviewer #1: The authors of this manuscript investigated the incidence of initiating pharmacological treatments for insomnia after COVID-19. I have several comments for this manuscript.

1. The authors need to cite appropriate previous studies and explain the scientific rationale on incidence of insomnia increase versus post-COVID-19 infection is so interesting. And then, authors need to support their study approach indicating the incidence of insomnia medications increase in individuals with post-COVID-19 infection can be reliable to measure that research question. However, the authors fail to explain the impact toward public health on COVID-19 infection and mental health which may be potentially a huge issue. These related statements remain fragile.

Thank you for your suggestions, as you indicated, we have changed the sentence as follows.

Page 5, Line80-100, Introduction, 2nd paragraph

The scientific rationale for examining insomnia incidence post-COVID-19 infection stems from several factors. First, the neuroinvasive potential of SARS-CoV-2 may directly affect brain regions involved in sleep regulation [5]. Second, the psychological stress, social isolation, and lifestyle changes associated with the pandemic can disrupt sleep patterns [6]. Third, persistent symptoms of long COVID, such as fatigue and breathing difficulties, may interfere with sleep quality [7]. Additionally, investigating insomnia medication use provides an objective measure of clinically significant sleep disturbances requiring pharmacological intervention.

2. NDB is a medical claim database. Claimed disease name in these medical insurance databases is often different from diagnosed disease. Seems related validation study is not available and/or prepared for this study topic. The authors need to be cautious on how the data became generated. The authors need to explain why this study approach can reasonably identify the study patients with insomnia and COVID-19 infection.

Thank you for your suggestions regarding the potential bias due to the insurance claimed database.

We have added the sentence in the limitation section.

Page 23, Line 462-465, Strength and Limitations section, 2nd paragraph, 4 and 5th sentences.

Fourth, the codes identified in this study have not undergone validation. In research that assesses the accuracy of Japanese diagnostic codes for hospitalized patients, sensitivity and specificity were 78.9% and 93.2%, respectively [37].

3. Study patients are only from 5 prefectures; however, using NDB dataset. Please provide information related to methodologies that other researchers can reproduce this result. Furthermore, are these populations representable of nationwide Japan? Why did you choose these 5 prefectures instead of 47? I think regions with higher incidence rate and prevalence of mental health diseases are missing from this study population.

Thank you for your suggestions regarding the selection bias due to regional issues. As you indicated, we have change the sentence as follows:

Page 23, Line 471-475, Discussion, Strength and Limitation, 8th sentence

Sixth, this study sampled only patients in the five prefectures, not in Japan as a whole. Although the population composition does not differ from that of Japan as a whole, the possibility cannot be denied that the risk of COVID-affected patients and the frequency of insomnia patients may differ from those in other prefectures (Supplementary Table4).

4. Table 1: Was the total study patients (N=4.4 million) reasonable level for explaining the incidence/ prevalence of insomnia among these 5 prefectures?

Thank you for your suggestions, regarding the study populations in this study. Since matching process might reduced the generalizability in this study, we have added the sentence in the limitation section.

Page 23, Line 468-471, Discussion, Strength and Limitations section, 2nd paragraph, 6th sentence.

Fifth, despite efforts to reduce confounding through matching on key variables, the study may still be subject to residual confounding from unmeasured factors not accounted for in the matching process, potentially biasing the results and limiting generalizability to broader populations.

5. The authors need to provide scientifically reasonable information on why the outcomes are defined appropriately in this study settings since claimed disease and actual diagnosed disease are often different.

6. The authors need to provide scientifically reasonable information on why the prescriptions were made only for insomnia in this study settings and made it possible to exclude other indications.

Thank you for pointing this out. In defining medication in this study, benzodiazepines and antidepressants indicated for anxiety and other conditions were not used as outcomes. We have added a description of this in the Outcome section. The results in the text have changed slightly because the matching was performed again accordingly.

Page 8, Line 159-167, Outcome section,1st and 2nd sentences.

The outcome of interest of this study was the initiation of prescription for insomnia which are available for outpatients in Japan, with insomnia being the main indication, and other indications such as anxiety and epilepsy being excluded. We categorize the drug as follows: Melatonin Receptor Agonist (MRA), Non-Benzodiazepine Hypnotic (non-BZO), benzodiazepine (short-acting [SA-BZO], intermediate-/long-acting [ILA-BZO]), Orexin Receptor Antagonist (ORA) and Barbiturates.

7. The authors calculated CCI scores based on claimed insurance disease which is different from actual diagnosed disease. Please cite appropriate previous studies and explain how this approach is scientifically reasonable and reliable.

Thank you for your suggestions regarding CCI scores on this study, as you indicated we have added the sentence in the limitation section.

Page 24, Line 469-472, Strength and Limitations section, 2nd paragraph, 6th sentences.

Although the validity of each disease was not determined, the CCI score calculated using the Japanese health insurance database was reported to be around 0.7-0.85 for c-statistics, which we believe indicates a certain validity of the background diseases in this study [38, 39].

Reviewer #2: Thank you for the opportunity to review the manuscript. I believe that several points need to be addressed.

Major issues:

1. The authors should address the potential for differential misclassification bias related to insomnia. It is plausible that the majority of patients do not receive treatment for insomnia, and individuals in the exposure group may be more likely to receive such treatment compared to those in the control group, potentially due to differences in help-seeking behavior.

I have added this to the Discussion section, as the impact as help seeking behavior you pointed out is indeed relevant.

Page 19, Line 370-376, Discussion section, 4th paragraph,

Furthermore, individuals with COVID-19 might be at an elevated risk of seeking medical attention compared to those without the disease, due to the diverse range of symptoms they exhibit [27]. Moreover, the public health emphasis on early detection and isolation of COVID-19 cases may contribute to this increased healthcare-seeking behavior [28]. Individuals experiencing symptoms consistent with COVID-19 may be more likely to reach out to healthcare providers for testing, diagnosis, and guidance on appropriate management and isolation measures [29].

2. The authors concluded that the initiation of insomnia medication highlights the necessity of mental health support. However, clinical guidelines for the treatment of insomnia generally recommend nonpharmacological interventions as the first-line approach. Therefore, the initiation of pharmacological treatment for insomnia does not necessarily indicate a need for mental health support, but may instead reflect potentially inappropriate prescribing practices.

In Japan, cognitive-behavioral therapy is not the primary treatment option, as it is in other countries, and is only recommended for certain patients according to the guidelines. This issue may be linked to the study's findings, which align with the projected increase in insomnia cases, and could be associated with the elevated risk of benzodiazepine dependence in Japan. We have inc

---

## [Decision Letter · Decision Letter 1]

18 May 2025

Dear Dr. Miyamori,

Thank you for submitting your manuscript to PLOS ONE. After careful consideration, we feel that it has merit but does not fully meet PLOS ONE’s publication criteria as it currently stands. Therefore, we invite you to submit a revised version of the manuscript that addresses the points raised during the review process.

**ACADEMIC EDITOR:**

The design of this study does not seem to be sufficient to achieve your original research objectives. This is partly related to the difficulties in conducting claim database research, and it would be preferable to conduct another study with a more improved design to achieve the original research objectives.

However, the results shown in the revised manuscript were considered to have potential value for publication with appropriate explanations.

Each reviewer has requested additional comments, mainly regarding the interpretation of the results and the explanation of limitations.

We look forward to receiving your revised manuscript.

Kind regards,

Tatsuya Noda

Academic Editor

PLOS ONE

Journal Requirements:

Reviewers' comments:

Reviewer's Responses to Questions

**Comments to the Author**

Reviewer #1: (No Response)

Reviewer #2: All comments have been addressed

2. Is the manuscript technically sound, and do the data support the conclusions?

Reviewer #1: Partly

Reviewer #2: Partly

3. Has the statistical analysis been performed appropriately and rigorously?

Reviewer #1: I Don't Know

Reviewer #2: Yes

4. Have the authors made all data underlying the findings in their manuscript fully available?

Reviewer #1: Yes

Reviewer #2: Yes

5. Is the manuscript presented in an intelligible fashion and written in standard English?

Reviewer #1: No

Reviewer #2: Yes

Reviewer #1: The authors responded to the raised comments though still not sufficient to improve the scientific sound.

1. The authors sticks on showing CCI scores which is NOT reliable enough. The authors failed to explain and justify how the results related to CCI can be implemented in clinical practice regarding their research question. For instance, CCI is often used for considering the conditions of frailty in clinical practice. It is unclear why the authors presented the results in this study on CCI which is NOT reliable enough. How does this result contribute to clinical practice? Its implementation remains totally vague.

2. Introduction is too long. Some statement is NOT directly related to the research question. The authors need to focus on appropriate citation to support their storyline with appropriate length.

3. Generalizability remains doubtful. The authors need to add statements related to external validity. Is the result consistent with previous results? What will be the perspectives of the study group, related to the interpretation of the results? Why is the statement related to the interpretation reasonable? These discussion remains fragile.

Reviewer #2: Thank you for the opportunity to review the manuscript. I believe that a minor issue need to be addressed.

Minor issues:

1. The authors should address the potential for differential misclassification bias related to insomnia. It is plausible that the majority of patients do not receive treatment for insomnia, and individuals in the exposure group may be more likely to receive such treatment compared to those in the control group, potentially due to differences in help-seeking behavior.

[Author’s response] I have added this to the Discussion section, as the impact as help seeking behavior you pointed out is indeed relevant.

Page 19, Line 370-376, Discussion section, 4th paragraph,

Furthermore, individuals with COVID-19 might be at an elevated risk of seeking medical attention compared to those without the disease, due to the diverse range of symptoms they exhibit [27]. Moreover, the public health emphasis on early detection and isolation of COVID-19 cases may contribute to this increased healthcare-seeking behavior [28]. Individuals experiencing symptoms consistent with COVID-19 may be more likely to reach out to healthcare providers for testing, diagnosis, and guidance on appropriate management and isolation measures [29].

[Additional comments] The authors did not address misclassification bias (i.e., the potential for bias to lead to distorted estimates), but rather discussed only potential sources of bias. Please clarify the likely direction and magnitude of the misclassification bias.

2. The authors concluded that the initiation of insomnia medication highlights the necessity of mental health support. However, clinical guidelines for the treatment of insomnia generally recommend nonpharmacological interventions as the first-line approach. Therefore, the initiation of pharmacological treatment for insomnia does not necessarily indicate a need for mental health support, but may instead reflect potentially inappropriate prescribing practices.

[Author’s response] In Japan, cognitive-behavioral therapy is not the primary treatment option, as it is in other countries, and is only recommended for certain patients according to the guidelines. This issue may be linked to the study's findings, which align with the projected increase in insomnia cases, and could be associated with the elevated risk of benzodiazepine dependence in Japan. We have incorporated this point into the Discussion section.

[Additional comments] Thank you for your revision. I have no further comments.

3. The authors stated that the rationale for this study was to investigate the long-term effects of COVID-19 on mental health. However, the median follow-up period was only seven months. The authors should clarify the justification for this follow-up duration in the context of evaluating long-term outcomes.

[Author’s response] Thank you. As you mentioned, the median time in this study was 7 months, which may not be considered a long period of time. Therefore, I have removed the ”long-term” part from the text.

[Additional comments] Thank you for your revision. I have no further comments.

4. The authors identified insomnia medications using ATC codes. For transparency and reproducibility, please provide a complete list of the included drug names, their routes of administration, and corresponding domestic drug codes. In particular, clarification is needed on the following points: What is 'lilmazepam'?; Why were 'rilmazafone' and 'melatonin' not included?; How was 'haloxazolam' identified, given that it does not have an ATC code?; How were 'midazolam' and 'haloxazolam' excluded from the ATC category NC05CD?; Was flunitrazepam injection included in the analysis?; Why were other insomnia medications, such as barbiturates, not considered?

[Authors’ response] Thank you for your observations. As you noted, we have rectified certain errors. Furthermore, we have not utilized injectable drugs at this stage and have included an additional note in the methodology section to reflect this.

[Additional comments] Thank you for your revision. I have no further comments.

**Do you want your identity to be public for this peer review?** For information about this choice, including consent withdrawal, please see our Privacy Policy

Reviewer #1: No

Reviewer #2: No

While revising your submission, please upload your figure files to the Preflight Analysis and Conversion Engine (PACE) digital diagnostic tool, https://pacev2.apexcovantage.com/. PACE helps ensure that figures meet PLOS requirements. To use PACE, you must first register as a user. Registration is free. Then, login and navigate to the UPLOAD tab, where you will find detailed instructions on how to use the tool. If you encounter any issues or have any questions when using PACE, please email PLOS at figures@plos.org

---

## [Author Response · Author response to Decision Letter 2]

24 Jun 2025

6. Review Comments to the Author

Reviewer #1: The authors responded to the raised comments though still not sufficient to improve the scientific sound.

1. The authors sticks on showing CCI scores which is NOT reliable enough. The authors failed to explain and justify how the results related to CCI can be implemented in clinical practice regarding their research question. For instance, CCI is often used for considering the conditions of frailty in clinical practice. It is unclear why the authors presented the results in this study on CCI which is NOT reliable enough. How does this result contribute to clinical practice? Its implementation remains totally vague.

Response: Thank you for your suggestions regarding the clinical implication of CCI in the clinical practice. We have added the sentences as you indicated.

Page 19, Line 303-307, Discussion section 5th paragraph, 1-3 sentences.

Critically, the NDB does not contain the functional, cognitive, or social variables required to construct formal frailty indices, nor the complete problem lists needed for standard multimorbidity counts. Consequently, we used CCI as a pragmatic surrogate, since higher CCI scores correlate well with frailty and multimorbidity [34, 35]. As in previous studies, there was an increase in both younger and older age groups.

Page 19-20, Line 315-321, Discussion section 5th paragraph, last 2 sentences.

These multimorbid, functionally vulnerable individuals account for the largest excess burden of post-COVID insomnia pharmacotherapy and are also most susceptible to hypnotic-related harms (falls, delirium, functional decline). Embedding brief sleep screening, fall-prevention counselling, and medication reviews into routine visits for this subgroup provides a pragmatic, resource-efficient way to translate our findings into clinical care, even where real-time CCI calculation is not feasible.

2. Introduction is too long. Some statement is NOT directly related to the research question. The authors need to focus on appropriate citation to support their storyline with appropriate length.

Thank you for your suggestions, as you indicated we have deleted redundant sentences and simplify the statement for aims in the Introduction section.

3. Generalizability remains doubtful. The authors need to add statements related to external validity. Is the result consistent with previous results? What will be the perspectives of the study group, related to the interpretation of the results? Why is the statement related to the interpretation reasonable? These discussion remains fragile.

Thank you for your suggestions, regarding the generalizability in our study. As you indicated we have added discussion section for the comparison with previous studies, and perspectives with interpretation of our study results.

Page 8, Line 116-119, Methods, Study participants, 3rd sentence.

In the five prefectures included in this study, the age-demographic composition was similar to that of the nation as a whole, and the cost frequency for psychiatric disorders as of 2021 did not show extreme bias. (Supplementary Tables 1 and 2).

Supplemental Table 2. Average per capita medical expenses related to mental illness in targeted prefectures

Page 9, Line 143-145, Methods, Data collection section, 2nd sentence; Japan's universal health insurance system features a detailed NDB that thoroughly records each outpatient appointment, hospital stay, diagnostic code, and medication prescribed for all residents.

Page 16, Line 255-164, Discussion section, 2nd paragraph

Previous studies have reported a sharp increase in hyponotics immediately after the COVID-19 epidemic, with an increase of 0.23 boxes per 1000 persons, [4] and a risk of being diagnosed with insomnia about 1.92 times higher six months after the illness and 2.46 times higher even one year later. [22] [23] It has also been reported that 30% of patients with severe PASC required sleep-inducing drugs for an extended period of time. [24]Using the Population database, which is a highly complete database, the present study observed a longer period of time than previous studies and found that the risk after COVID-19 disease persisted for more than 2 years.Hypnotics, medications generally used to treat insomnia, may be more frequently prescribed to manage persistent sleep-related symptoms in these patients.

Page 17,Line 272-275, Discussion section, 3rd paragraph

Based on our findings, in the context of Japan's unique clinical approach, [26] it is assumed that our results reflect the number of new pharmacotherapies for insomnia. Conversely, it is important to consider that the variations in treatment interventions for sleeping pills in Japan may limit the generalizability of this study.

Page 17-18, Line 277-286, Discussion section, 4th paragraph, 1-5th sentences.

No previous studies have examined which sleeping medications are more likely to be used for insomnia after COVID-19. The secondary endpoint showed a high rate of increase in the non-BZO and MRA categories, while a certain number of BZOs were also prescribed..The amount of BZO and non-BZO prescriptions in existing patients in recent years is reported to be 60% and 37%, indicating that the frequency of BZO prescriptions is still high in Japan.[27] Historical Japanese prescription patterns for new sleep medications using NDB from 2010-2019 showed that BZOs accounted for 30-50% of prescriptions, ORAs for 20%, and MRAs for about 3-6% of prescriptions[28]。In the current study, ORAs accounted for about 40% of the total, and MRAs were also increasing in prescription volume, at 10%. Japan has a high prevalence of benzodiazepine abuse disorders[29].

Page 18, Line 290-293, Discussion section, 4th paragraph, last two sentences.

The results of this study suggest that the frequency of BZO prescriptions may be decreasing compared to the past. However, the high incidence of benzodiazepine dependence in Japan should continue to be monitored, which necessitates future improvement.

Page 23, Line 378-382, Discussion section, last paragraph, 4th and 5th last sentences.

Sixth, this study sampled only patients in the five prefectures, not in Japan as a whole, and the possibility cannot be denied that the risk of COVID-affected patients and the frequency of insomnia patients may differ from those in other prefectures. Similarly, generalizability is limited in foreign countries with different cultural backgrounds and treatment policies.

Reviewer #2: Thank you for the opportunity to review the manuscript. I believe that a minor issue need to be addressed.

Minor issues:

1. The authors should address the potential for differential misclassification bias related to insomnia. It is plausible that the majority of patients do not receive treatment for insomnia, and individuals in the exposure group may be more likely to receive such treatment compared to those in the control group, potentially due to differences in help-seeking behavior.

[Author’s response] I have added this to the Discussion section, as the impact as help seeking behavior you pointed out is indeed relevant.

Page 19, Line 370-376, Discussion section, 4th paragraph,

Furthermore, individuals with COVID-19 might be at an elevated risk of seeking medical attention compared to those without the disease, due to the diverse range of symptoms they exhibit [27]. Moreover, the public health emphasis on early detection and isolation of COVID-19 cases may contribute to this increased healthcare-seeking behavior [28]. Individuals experiencing symptoms consistent with COVID-19 may be more likely to reach out to healthcare providers for testing, diagnosis, and guidance on appropriate management and isolation measures [29].

[Additional comments] The authors did not address misclassification bias (i.e., the potential for bias to lead to distorted estimates), but rather discussed only potential sources of bias. Please clarify the likely direction and magnitude of the misclassification bias.

Thank you for your suggestions regarding the potential misclassification in our study. We have added the sentence as follows.

Page 22, Discussion, 2nd last paragraph, 2nd and 4th sentences.

Second, selection bias may exist due to misclassification of COVID-19 infection status and the outcome status.

Conversely, insomnia may be more frequently prescribed as a health-seeking behavior in patients affected by post-COVID-19 conditions, potentially amplifying the observed results.

---

## [Decision Letter · Decision Letter 2]

3 Aug 2025

Dear Dr. Miyamori,

Thank you for submitting your manuscript to PLOS ONE. After careful consideration, we feel that it has merit but does not fully meet PLOS ONE’s publication criteria as it currently stands. Therefore, we invite you to submit a revised version of the manuscript that addresses the points raised during the review process.

We look forward to receiving your revised manuscript.

Kind regards,

Tatsuya Noda

Academic Editor

PLOS ONE

Journal Requirements:

Additional Editor Comments:

First, I deeply apologize for the several weeks delay in responding to your submission. I was unable to notice your resubmission due to a delay in checking my email address, as it had been recently changed, and the complexity of the PLOS ONE editing system.

As pointed by Reviewer 1, there are several additional descriptions that lack sufficient supporting evidence or are inconsistent with the purpose of the addition. Please consider revising your manuscript to address these issues and resubmit it.

Best regards,

Reviewers' comments:

Reviewer's Responses to Questions

**Comments to the Author**

Reviewer #1: All comments have been addressed

Reviewer #2: All comments have been addressed

2. Is the manuscript technically sound, and do the data support the conclusions?

Reviewer #1: Partly

Reviewer #2: Yes

3. Has the statistical analysis been performed appropriately and rigorously?

Reviewer #1: I Don't Know

Reviewer #2: Yes

4. Have the authors made all data underlying the findings in their manuscript fully available?

Reviewer #1: Yes

Reviewer #2: No

5. Is the manuscript presented in an intelligible fashion and written in standard English?

Reviewer #1: Yes

Reviewer #2: Yes

Reviewer #1: The authors responded to the raised comments though still insufficient.

1.The authors added statements on CCI scores and fraility and comorbidities. However, the statement remains general and vague, no explanation specific to the research question defined by the authors. The authors failed to explain and justify how the results related to CCI can be implemented in clinical practice regarding their research question. The authors need to explain and speculate how CCI score can be used in clinical practice for insomnia and COVID-19 therapy related to the main result. Presenting the result without any potential implementation to the clinical practice is worthless. Why did you choose NDB and present "CCI" like score in this manuscript? If there is no reason and explanation for this, this data is not interesting and contributing to improve current clinical practice. Please explore the implications of your finding appropriately.

2. The authors need to avoid unqualified statements and conclusions not adequately supported by the data. Some statement remains NOT directly related to the research question. The additional statements made by the authors are mostly not supported by appropriate citations. The authors need to learn how to prepare a manuscript following basics in medical writing. Please improve your statements by referring to ICJME website (https://www.icmje.org/recommendations/browse/manuscript-preparation/preparing-for-submission.html). Discussion section is too long and need to downsize by 50-70%. However, Discussion section lacks statement for exploring the implications of your findings.

3. The authors added statements though mostly NOT supported by appropriate citations (e.g., lines 116-119, lines 262-264, 290-293,315-321). Please improve.

4. Cite appropriate clinical practice guideline for statement in line 266.

5. Are statements in lines 274-286 and lines 303-307 necessary for this research question? The linkage between this statement and the goal of this study remain totally unclear.

Reviewer #2: Thank you for the revised manuscript. I have no additional comments and commend the authors for their efforts.

**Do you want your identity to be public for this peer review?** For information about this choice, including consent withdrawal, please see our Privacy Policy

Reviewer #1: No

Reviewer #2: No

While revising your submission, please upload your figure files to the Preflight Analysis and Conversion Engine (PACE) digital diagnostic tool, https://pacev2.apexcovantage.com/. PACE helps ensure that figures meet PLOS requirements. To use PACE, you must first register as a user. Registration is free. Then, login and navigate to the UPLOAD tab, where you will find detailed instructions on how to use the tool. If you encounter any issues or have any questions when using PACE, please email PLOS at figures@plos.org

---

## [Author Response · Author response to Decision Letter 3]

15 Sep 2025

We sincerely thank the reviewers and editors for their careful reading of our manuscript and for their constructive feedback. Below we provide a point-by-point response. Each reviewer comment is followed by our response and the corresponding manuscript changes .

Reviewer #1

Comment 1

Reviewer comment: The authors added statements on CCI scores and frailty and comorbidities. However, the statement remains general and vague… Please explain and speculate how CCI score can be used in clinical practice for insomnia and COVID-19 therapy.

Response: We thank the reviewer for this important point. We clarified the clinical implications of CCI, emphasizing its link to multimorbidity and the need for proactive sleep screening and anticipatory guidance in multimorbid patients after COVID-19.

Change in manuscript (Discussion, Paragraph 3 (line 265 ~279)):

Before:

Critically, the NDB does not contain the functional, cognitive, or social variables required to construct formal frailty indices, nor the complete problem lists needed for standard multimorbidity counts. Consequently, we used CCI as a pragmatic surrogate, since higher CCI scores correlate well with frailty and multimorbidity [26, 27]. As in previous studies, there was an increase in both younger and older age groups. In the subgroup analysis, it was observed that prescriptions for sleep medication were approximately 1.5 times more likely to be issued to younger individuals. The use of hypnotics in children is associated with potential adverse effects, such as suppression of the hypothalamic-gonadal axis [46], which can trigger exacerbations of headaches, hallucinations, and dizziness [47]. Similarly, an increased risk of prescription was found in older individuals with higher CCI. In these groups, it is assumed that pre-existing comorbidities of chronic health conditions, cognitive decline, drug-drug interaction due to polypharmacy, and risk of social isolation might exacerbate and lead to an increased risk of insomnia [47]. These multimorbid, functionally vulnerable individuals account for the largest excess burden of post-COVID insomnia pharmacotherapy and are also most susceptible to hypnotic-related harms (falls, delirium, functional decline). Embedding brief sleep screening, fall-prevention counselling, and medication reviews into routine visits for this subgroup provides a pragmatic, resource-efficient way to translate our findings into clinical care, even where real-time CCI calculation is not feasible.

After:

Patients with higher CCI scores showed an increased risk of initiating insomnia pharmacotherapy after COVID-19. Because the CCI is closely related to multimorbidity, [26, 27] this finding indicates that individuals with multiple comorbid conditions are not only vulnerable to exacerbations of their underlying diseases after COVID-19 but also at elevated risk of developing insomnia. In routine outpatient practice, when multimorbid patients contract COVID-19, it is important to provide anticipatory guidance based on this risk and to incorporate follow-up assessments for Long COVID manifestations, including sleep disturbances, into ongoing care. While attention in these patients frequently centers on respiratory or cardiovascular issues, our findings suggest that clinicians should also consider sleep, a domain intricately connected to quality of life. [28] Engaging in proactive questioning, such as inquiring, “Have you been sleeping well since your COVID-19 infection?” may assist in identifying sleep disturbances that patients might be hesitant to report voluntarily. This method could potentially decrease the number of undiagnosed cases of post-COVID insomnia and promote earlier intervention.

Comment 2

Reviewer comment: The authors need to avoid unqualified statements and conclusions not adequately supported by the data. Discussion section is too long and needs to be downsized by 50–70%. However, it lacks statement for exploring the implications.

Response: We shortened the Discussion by removing redundant text and speculative sections, while strengthening the focus on clinical implications.

Change in manuscript (Discussion, Paragraph 2 (lines 256–260)):

Before:

Previous studies have reported a sharp increase in hyponotics immediately after the COVID-19 epidemic, with an increase of 0.23 boxes per 1000 persons, [4] and a risk of being diagnosed with insomnia about 1.92 times higher six months after the illness and 2.46 times higher even one year later. [23] [24] It has also been reported that 30% of patients with severe PASC required sleep-inducing drugs for an extended period of time. [25]Using the Population database, which is a highly complete database, the present study observed a longer period of time than previous studies and found that the risk after COVID-19 disease persisted for more than 2 years.Hypnotics, medications generally used to treat insomnia, may be more frequently prescribed to manage persistent sleep-related symptoms in these patients.

After:

Studies reported a sharp rise in hyponotics after COVID-19, with an increase of 0.23 boxes per 1000 persons,[4]and higher insomnia diagnosis risk of 1.92 times at six months and 2.46 times at one year.[23, 24]Additionally, 30% of severe PASC patients needed sleep-inducing drugs long-term.[25]Using the Population database, this study found the risk after COVID-19 persisted beyond 2 years.

Change in manuscript (Discussion, Paragraph 2 (lines 260–263)):

Before:

In Japan, sleep inducing drugs are the first choice of treatment for insomnia.n Japan, cognitive-behavioral therapy for sleep disorder is not covered by insurance and is not reimbursed. In addition, the 2014 Japanese Practical Guidelines for Insomnia do not recommend cognitive-behavioral therapy as a first line therapy, and consider treatment when there is little improvement with pharmacotherapy　[26]. In fact, reports indicate that only 60,000 patients nationwide received CBT for mood disorders between 2010 and 2015 [14]. Based on our findings, in the context of Japan's unique clinical approach, [27] it is assumed that our results reflect the number of new pharmacotherapies for insomnia. Conversely, it is important to consider that the variations in treatment interventions for sleeping pills in Japan may limit the generalizability of this study.

After:

In Japan, cognitive-behavioral therapy for insomnia (CBT-I) is not covered by insurance, and pharmacotherapy remains the predominant treatment. [13] Therefore, the prescription of hypnotics in this study may reflects the onset of new cases of insomnia.

Change in manuscript (Discussion):

Before:

No previous studies have examined which sleeping medications are more likely to be used for insomnia after COVID-19. The secondary endpoint showed a high rate of increase in the non-BZO and MRA categories, while a certain number of BZOs were also prescribed. The amount of BZO and non-BZO prescriptions in existing patients in recent years is reported to be 60% and 37%, indicating that the frequency of BZO prescriptions is still high in Japan.[28] Historical Japanese prescription patterns for new sleep medications using NDB from 2010-2019 showed that BZOs accounted for 30-50% of prescriptions, ORAs for 20%, and MRAs for about 3-6% of prescriptions[29].In the current study, ORAs accounted for about 40% of the total, and MRAs were also increasing in prescription volume, at 10%. Japan has a high prevalence of benzodiazepine abuse disorders[30] and has attempted to align with WHO guidelines on benzodiazepine dependence:[31] to limit the duration of prescriptions, implement awareness campaigns, educate about the dangers of long-term BZO use, and strengthen monitoring and surveillance. The results of this study suggest that the frequency of BZO prescriptions may be decreasing compared to the past. However, the high incidence of benzodiazepine dependence in Japan should continue to be monitored, which necessitates future improvement.

After:

(Deleted)

Change in manuscript (Discussion, Strength and limitation 314-321):

Before:

Furthermore, individuals with COVID-19 might be at an elevated risk of seeking medical attention compared to those without the disease, due to the diverse range of symptoms they exhibit [32]. Moreover, the public health emphasis on early detection and isolation of COVID-19 cases may contribute to this increased healthcare-seeking behavior [33]. Individuals experiencing symptoms consistent with COVID-19 may be more likely to reach out to healthcare providers for testing, diagnosis, and guidance on appropriate management and isolation measures [34].

After: Added in the limitation section

A potential limitation of this study is the increased likelihood of healthcare-seeking behavior among individuals with COVID-19 compared to those without the disease. [40, 41] The emphasis on early detection and isolation in public health messaging may have led to higher rates of medical attention-seeking. [41] These factors could have influenced the observed patterns in healthcare utilization and should be considered when interpreting the results.

Change in manuscript (Discussion, Line 281-290, Discussion 4th paragraph):

Before:

Insomnia poses a threat of exacerbating several physical and psychiatric disorders. It is accompanied by psychiatric and psychological comorbidities such as anxiety, depression, and heightened daytime stress reactivity. [39] Furthermore, chronic insomnia is reportedly associated with a decline in cognitive function, a decrease in quality of life, and an increase in the likelihood of injuries, suicidal ideation, cardiovascular risk, diabetes, and cancer. [40-43] The COVID-19 pandemic posed significant challenges that could lead to negative consequences for all generations, including an increased burden on healthcare providers. Additionally, the pandemic resulted in economic losses owing to reduced productivity. [44] To mitigate these effects and protect the health of affected populations, appropriate post-infection surveillance and the introduction of therapeutic interventions, including cognitive behavioral therapy, are urgently needed. Conversely, while cognitive behavioral therapy is the preferred treatment internationally, the rising use of sleeping pills in Japan may indicate a current demand for psychological support. However, it may also suggest an increase in the incidence of inappropriate prescribing, as previously mentioned.

The mechanism of insomnia in individuals with PASC is complex and involves multiple biological and psychological factors., [21, 45, 46] Although the precise mechanism remains unclear, several potential pathways have been identified. Insomnia in patients with PASC likely results from a combination of neuroinflammatory processes, autonomic dysfunction, direct viral effects on the central nervous system, psychological stressors, circadian rhythm disruptions, and physical symptoms. [47] The multifactorial nature of insomnia in COVID highlights the necessity for a comprehensive approach to treatment that addresses both physiological and psychological components. For any risk, it is crucial to anticipate effects and link them to regular monitoring and evaluation after the disease has occurred, leading to early identification and intervention.

After:

Insomnia exacerbates physical and psychiatric disorders, often with anxiety, depression, and increased stress. Chronic insomnia is linked to cognitive decline, reduced quality of life, and higher risks of injuries, suicidal thoughts, cardiovascular issues, diabetes, and cancer. [29-32] In addition, the mechanism of insomnia in PASC involves multiple biological and psychological factors. [21, 33, 34] Insomnia in PASC patients likely results from neuroinflammatory processes, autonomic dysfunction, direct viral effects on the central nervous system, psychological stressors, circadian rhythm disruptions, and physical symptoms .[35] The COVID-19 pandemic heightened challenges, impacting all generations and increasing healthcare burdens and economic losses due to reduced productivity. [36]

Strength and Limitation section (Line 321-324)

Before:

Sixth, this study sampled only patients in the five prefectures, not in Japan as a whole, and the possibility cannot be denied that the risk of COVID-affected patients and the frequency of insomnia patients may differ from those in other prefectures. Similarly, generalizability is limited in foreign countries with different cultural backgrounds and treatment policies.

After:

Sixth, this study sampled patients in only five prefectures, not all of Japan, and COVID-affected and insomnia patient rates may differ across prefectures. Similarly, generalizability is limited in countries with different cultural backgrounds and treatment policies.

Comment 3

Reviewer comment: Some statements are not supported by citations (e.g., lines 116–119, 262–264, 290–293, 315–321).

Response: We corrected and updated references. For example, we replaced the old guideline citation with Takaesu et al. (2023) and replaced the benzodiazepine discontinuation statement with Shimane et al. (2015). We also ensured speculative statements were removed or revised.

Line 116-119: In the five prefectures included in this study, the age-demographic composition was similar to that of the nation as a whole, and the cost frequency for psychiatric disorders as of 2021 did not show extreme bias [19] (Supplementary Tables 1 and 2).

Before: 19. Minisitry of Health Labour and Welfare. Analysis of Regional Differences in Medical Expenses 2021 [cited 2025 5/25/2025]. Available from: https://www.mhlw.go.jp/stf/seisakunitsuite/bunya/kenkou_iryou/iryouhoken/database/iryomap/index.html.

After: 19. Minisitry of Health Labour and Welfare. Database on regional disparities in medical expenditure (Iryohi map) 2023 [cited 2025 5/25/2025]. Available from: https://www.mhlw.go.jp/content/iryohi_r04_kiso.xlsx (Sheet 5: Per capita age-adjusted medical expenses by disease classification, Kokuho and Kokikoreisha insurance).

Line 262-264: We have deleted the sentence below

Hypnotics, medications generally used to treat insomnia, may be more frequently prescribed to manage persistent sleep-related symptoms in these patients.

Line 290-293: We have deleted the sentence below

The results of this study suggest that the frequency of BZO prescriptions may be decreasing compared to the past. However, the high incidence of benzodiazepine dependence in Japan should continue to be monitored, which necessitates future improvement.

Line 315-321: We have deleted the sentence below

These multimorbid, functionally vulnerable individuals account for the largest excess burden of post-COVID insomnia pharmacotherapy and are also most susceptible to hypnotic-related harms (falls, delirium, functional decline). Embedding brief sleep screening, fall-prevention counselling, and medication reviews into routine visits for this subgroup provides a pragmatic, resource-efficient way to translate our findings into clinical care, even where real-time CCI calculation is not feasible.

Comment 4

Reviewer comment: Cite appropriate clinical practice guideline for statement in line 266.

Response: We agree and change the reference to the appropriate one.

Change in manuscript (Discussion, Paragraph 5 (line 260-262)):

Before:

In Japan, cognitive-behavioral therapy for sleep disorder is not covered by insurance and is not reimbursed. In addition, the 2014 Japanese Practical Guidelines for Insomnia do not recommend cognitive-behavioral therapy as a first line therapy, and consider treatment when there is little improvement with pharmacotherapy [26].

After:

In Japan, cognitive-behavioral therapy for insomnia (CBT-I) is not covered by insurance, and pharmacotherapy remains the predominant treatment. [13]

Comment 5

Reviewer comment: Are statements in lines 274–286 and 303–307 necessary? The linkage with the study’s goal is unclear.

Response: We agree and removed these sections as they were speculative or unrelated to the research question.

Change in manuscript (Discussion, Paragraph 6 (lines 274–286)):

Before:

No previous studies have examined which sleeping medications are more likely to be used for insomnia after COVID-19. The secondary

---

## [Decision Letter · Decision Letter 3]

26 Nov 2025

Dear Dr. Miyamori,

Thank you for submitting your manuscript to PLOS ONE. After careful consideration, we feel that it has merit but does not fully meet PLOS ONE’s publication criteria as it currently stands. Therefore, we invite you to submit a revised version of the manuscript that addresses the points raised during the review process.

We look forward to receiving your revised manuscript.

Kind regards,

Armaan Jamal

Guest Editor

PLOS ONE

Journal Requirements:

Reviewers' comments:

Reviewer's Responses to Questions

**Comments to the Author**

Reviewer #1: All comments have been addressed

Reviewer #2: All comments have been addressed

2. Is the manuscript technically sound, and do the data support the conclusions?

Reviewer #1: Partly

Reviewer #2: Yes

3. Has the statistical analysis been performed appropriately and rigorously?

Reviewer #1: I Don't Know

Reviewer #2: Yes

4. Have the authors made all data underlying the findings in their manuscript fully available?

Reviewer #1: Yes

Reviewer #2: No

5. Is the manuscript presented in an intelligible fashion and written in standard English?

Reviewer #1: Yes

Reviewer #2: Yes

Reviewer #1: The authors responded to the raised comments though still insufficient. The authors added statements on CCI scores and fraility and comorbidities. The authors need to laser focus on strengthening what is interesting in the study findings. "The incidence of initiating insomnia medications increased approximately 1.7 times in individuals with post-COVID-19 infection compared with those without infections" will be the main findings of the study and this part remains interesting. However, the authors presented that older age and higher CCI scores were associated with the incidence of new insomnia medication use and this part remains NOT interesting. It is common sense that younger people have lower CCI with less medication use and elderly people have higher CCI and increased medication use in most population. What is different from this common sense? The authors need to specify on the difference of the study population and what is worthy to apply on clinical practice. The authors referred to the screening of insomnia initiation though the study result can be interpreted as ordinally implication that "elderly people have higher insomnia incidence" which is totally NOT interesting. The author needs to emphasize what is new and interesting related to the potential insomnia incidence for presenting the study result. The authors need to specify the statement, and specific explanation which relates to the research question defined by the authors themselves. Therefore, authors still failed to explain and justify how the results related to CCI can be implemented in the clinical practice regarding their research question. Please explain and speculate how this CCI score can be used in clinical practice for insomnia and COVID-19 therapy related to the main result. Presenting the result without any potential implementation to the clinical practice is worthless. The question "Why did you present CCI in this manuscript?" remains not answered yet. If there is no reason and explanation for this, the data related to CCI is redundant, not interesting, should be removed since it is not contributing to improve current clinical practice. Please explore the implications of your finding appropriately.

Reviewer #2: Thank you for your thorough revision and detailed responses to the previous comments. The authors have clearly addressed all major concerns raised in the earlier review round, and the manuscript has improved substantially in clarity, rigor, and interpretability.

**Do you want your identity to be public for this peer review?** For information about this choice, including consent withdrawal, please see our Privacy Policy

Reviewer #1: No

Reviewer #2: No

---

## [Author Response · Author response to Decision Letter 4]

26 Dec 2025

Reviewer #1 Comment:

The question "Why did you present CCI in this manuscript?" remains not answered yet…

Response

We sincerely thank the reviewer for this critical and constructive comment, which prompted us to reconsider the role and interpretation of the Charlson Comorbidity Index (CCI) in our manuscript.

We fully agree that higher CCI scores and older age are well-established correlates of increased medication use in general, and that presenting CCI as an independent risk factor for insomnia initiation does not constitute a novel or clinically actionable finding in itself. In this respect, we acknowledge that our previous presentation did not sufficiently clarify the rationale for including CCI-related results, nor did it adequately align with the primary research question of this study.

The primary purpose of incorporating CCI in this study was methodological rather than inferential. Specifically, CCI was used as a matching variable to balance baseline comorbidity burden between the COVID-19 and non-COVID-19 cohorts, thereby reducing confounding and isolating the effect of COVID-19 infection on subsequent initiation of insomnia pharmacotherapy. This design choice was essential to ensure internal validity, particularly given the strong association between multimorbidity and healthcare utilization.

Upon careful reconsideration of the reviewer’s concern, we concluded that presenting CCI-based subgroup analyses and discussing CCI as a clinical risk factor could distract from the central and novel message of this study—namely, that COVID-19 infection is associated with a substantially increased risk of initiating insomnia pharmacotherapy across all age groups, including children, adolescents, and younger adults, independent of baseline comorbidity burden.

In addition, in response to the reviewer’s concern that age-related findings may be interpreted as a reiteration of common clinical knowledge, we have removed all discussions that framed older age itself as a clinically meaningful or noteworthy risk factor for the initiation of insomnia pharmacotherapy.

Accordingly, we have revised the manuscript as follows: We removed all statements interpreting higher CCI scores as an independent risk factor for insomnia medication initiation. Furthermore, we deleted statements suggesting that older individuals were at higher risk per se, as such interpretations do not align with the primary objective of this study and may obscure the novel contribution of our findings.

The following CCI and older individuals-related interpretations have been removed:

・Abstract: references to increased risk among older individuals (Page 4, Lines 53–54).

・Methods: subgroup analysis for CCI categories (Page 11, Lines 170-171)

・Results: descriptive or interpretative statements emphasizing higher incidence in older age groups (Page 13, Lines 201; Page 14, Line 221-222).

・Discussion: age-centered interpretations implying that insomnia risk after COVID-19 is predominantly driven by older age (Page 15, Lines 240–242 and 246; Page 17, Line 286).

CCI is now explicitly described only as a covariate used for matching and confounding control, without further clinical interpretation.

By doing so, we sharpened the focus of the manuscript on its key and clinically relevant finding: that post-COVID-19 insomnia requiring pharmacological treatment is not limited to traditionally “high-risk” populations (e.g., older or multimorbid individuals), is also evident in younger and otherwise healthier populations. This has important implications for clinical practice, as it supports the need for vigilance and proactive screening for sleep disturbances in all patients following COVID-19, rather than targeting only those with advanced age or multiple comorbidities.

We believe that this revision directly addresses the reviewer’s concern and substantially strengthens the coherence, novelty, and clinical relevance of the manuscript.

In addition, in response to the reviewer’s concern that age-related findings may be interpreted as a reiteration of common clinical knowledge, we have removed all discussions that framed older age itself as a clinically meaningful or noteworthy risk factor for the initiation of insomnia pharmacotherapy.

---

## [Editor Report · Decision Letter 4]

7 Jan 2026

Impact of COVID-19 on New Pharmacotherapy for Insomnia: A Matched Cohort Study using the National Insurance Claims Database in Japan

PONE-D-25-01134R4

Dear Dr. Miyamori,

We’re pleased to inform you that your manuscript has been judged scientifically suitable for publication and will be formally accepted for publication once it meets all outstanding technical requirements.

Kind regards,

Armaan Jamal

Guest Editor

PLOS One

---

## [Editor Report · Acceptance letter]

PONE-D-25-01134R4

PLOS One

Dear Dr. Miyamori,

I'm pleased to inform you that your manuscript has been deemed suitable for publication in PLOS One. Congratulations! Your manuscript is now being handed over to our production team.

Kind regards,

on behalf of

Mr. Armaan Jamal

Guest Editor

PLOS One